# Disabled 1 Is Part of a Signaling Pathway Activated by Epidermal Growth Factor Receptor

**DOI:** 10.3390/ijms22041745

**Published:** 2021-02-09

**Authors:** Paula Dlugosz, Magdalena Teufl, Maximilian Schwab, Katharina Eva Kohl, Johannes Nimpf

**Affiliations:** Max Perutz Laboratories, Department of Medical Biochemistry, Medical University Vienna, 1030 Vienna, Austria; paula.dlugosz1@gmail.com (P.D.); Magdalenateufl@gmail.com (M.T.); mce.schwab@gmail.com (M.S.); katharina.eva.kohl@gmail.com (K.E.K.)

**Keywords:** Dab1, EGFR, EGF, cell proliferation, intestinal cells

## Abstract

Disabled 1 (Dab1) is an adapter protein for very low density lipoprotein receptor (VLDLR) and apolipoprotein E receptor 2 (ApoER2) and an integral component of the Reelin pathway which orchestrates neuronal layering during embryonic brain development. Activation of Dab1 is induced by binding of Reelin to ApoER2 and VLDLR and phosphorylation of Dab1 mediated by Src family kinases. Here we show that Dab1 also acts as an adaptor for epidermal growth factor receptor (EGFR) and can be phosphorylated by epidermal growth factor (EGF) binding to EGFR. Phosphorylation of Dab1 depends on the kinase activity of EGFR constituting a signal pathway independent of Reelin and its receptors.

## 1. Introduction

Disabled 1 (Dab1) was discovered in mice as intracellular adapter protein which plays a key function in a signal transduction pathway involved in the formation of neural networks and neuronal positioning in the developing brain [1,2]. The core event of this pathway is the “canonical” Reelin pathway reviewed in [3,4]. Reelin binds to the low-density lipoprotein receptors, apolipoprotein E receptor 2 (ApoER2) and very low-density lipoprotein receptor (VLDLR) [5], which leads to phosphorylation of Dab1 [6] which binds to the NPxY motifs on the intracellular domains of ApoER2 and VLDLR [7]. Phosphorylation of Dab1 is mediated by the Src family kinases Fyn and Src on specific tyrosine residues [8,9]. As recently demonstrated, the molecular event responsible for phosphorylation of Dab1 is receptor clustering by Reelin [10,11]. Phosphorylated Dab1 recruits downstream effectors such as Crk and phosphatidylinositide 3-kinase (PI3K), resulting in activation of several signaling cascades notably activating PKB/Akt and the CrkL/C3G/Rap1 pathway [12,13,14]. The later pathway was shown to regulate plasma membrane localization of N-cadherin [15] and is involved in polarization of migrating cortical neurons [16]. In addition, a crosstalk exists between Reelin signaling and the Notch pathway by Dab1 mediated inhibition of the degradation of the intracellular domain (ICD) of Notch [17]. Disruption of Reelin pathway leads to the well described “Reeler” phenotype in mice which is characterized by motor deficits and an abnormal layering of laminated structures in many regions of the brain (reviewed in [18,19]).

Besides this well-established role in the canonical Reelin pathway, Dab1 has a cell autonomous function in endothelial cells for vascularization during brain development. In this situation, Dab1 becomes phosphorylated by VEGF binding to VEGFR2 which interacts with ApoER2 on the cell surface establishing a crosstalk between both pathways [20].

In addition, a large body of evidence emerged recently, that the Reelin pathway or components thereof have important roles in non-neuronal tissues as well (reviewed in [21]). The small intestine is one example where Dab1 and VLDLR are expressed in the crypt-villus axis, whereas Reelin is expressed in myofibroblasts located beneath the intestinal epithelium suggesting that the Reelin pathway might be involved in cellular homeostasis of the rapid turnover of epithelial cells of the small intestine [22]. Along this line, Dab1 seems to play a role in tumor growth and metastatic progression [23] and is part of a Notch-Dab1-ABL axis which promotes colorectal cancer invasion and metastasis [24].

Recent characterization of pediatric T-lymphoblastic leukemia revealed that these cells are most similar to immature thymic precursors and exhibit strong overexpression of Dab1 as a common feature [25].

Here we demonstrate that Dab1 is part of a Reelin independent pathway by interaction with epidermal growth factor receptor (EGFR). In this combination Dab1 phosphorylation can be achieved by epidermal growth factor (EGF) and could be involved in organizing cell renewal and maintaining epithelial integrity of the intestine.

## 2. Results

Since Dab1 may act independently of Reelin and more importantly of ApoER2 and VLDLR we tested the possibility whether Dab1 might interact with other receptors involved in cell migration and proliferation and compared the sequences of the intracellular domains of a variety of receptor tyrosine kinase with those from ApoER2 and VLDLR. The consensus binding site for Dab1 (NPxY) mediating the interaction with ApoER2 and VLDLR [5,26] is present three times in the intracellular domain of EGFR (one of them is presented in Figure 1A in comparison with the respective sites in human and murine ApoER2 and VLDLR). Thus, we set out to explore the possibility whether Dab1 binds to EGFR. HEK293 cells were transiently transfected with EGFR and Dab1 and co-immunoprecipitation experiments were performed. As demonstrated in Figure 1B (Lanes 4–6), Dab1 is precipitated with an EGFR–specific antibody. As control experiment precipitation of Dab1 was carried out with an ApoER2–specific antibody (Lanes 1–3) demonstrating the well-established interaction of ApoER2 and Dab1 [7].

To verify the interaction between EGFR and Dab1 in a physiological context we used cell lysates derived from embryonic murine neurons and immunoprecipitations were performed as described for Figure 1B. First, we tested for the presence of all components and Western blots using specific antibodies against Dab1, ApoER2, and EGFR were performed (Figure 1C, Lanes 1 and 4; input). As expected, precipitation of ApoER2 led to the co-precipitation of Dab1 (Lane 2). As demonstrated in Lane 5, precipitation of EGFR also co-precipitated a small amount of Dab1 present in the lysate. The nature of the strong band migrating above Dab1 (Lane 5) is not clear yet, inasmuch as it cannot be detected in the input. This band is also visible when Dab1 was precipitated with ApoER2 from neuronal lysates (Figure 1C, Lane 2) and with EGFR from HEK293 lysates (Figure 1B, Lane 5). As control, unspecific IgGs were used, and neither ApoER2, EGFR, nor Dab1 was found in the precipitates (Lanes 3 and 6).

To test for co-localization of EGFR and Dab1 we expressed both proteins tagged to fluorophores (EGFR-mCherry; Dab1-mGFP) and assessed their expression pattern by confocal microscopy (Figure 1D). EGFR nicely decorates the entire cell membrane (Figure 1D, a); Dab1, in addition of being present at the cell membrane, is also located in the cytoplasm (b). The overlay of a and b (c) demonstrates that the fraction of Dab1 present at the cell membrane co-localizes with the receptor. Control experiments using mGFP (e,h) or mCherry (g,m) prove that the effect observed is not due to the used fluorescent tags. In addition, co-expression of EGFR and Dab1 was compared to the well described situation when Dab1 is expressed in the presence of ApoER2. As demonstrated in (j–l) presence of ApoER2 promotes translocation of Dab1 to the membrane, which in the absence of ApoER2 is mostly present in the cytosol (n). The same behavior is evident when EGFR is present instead of ApoER2 demonstrating that the co-localization of Dab1 with EGFR is comparable to that with ApoER2. In the absence of either ApoER2 or EGFR Dab1 remains in the cytosol (m–o).

Having demonstrated that Dab1 interacts with EGFR we tested whether this interaction might be part of a hitherto unknown signalling pathway. HEK293 cells expressing Dab1 and EGFR were treated for different time periods with EGF (10 ng/mL) and phosphorylated Dab1 and phosphorylated EGFR were visualized by immunoprecipitation and Western blotting using specific antibodies. As expected, addition of EGF led to rapid phosphorylation of EGFR (Figure 2A, Panel 3). EGFR phosphorylation reached maximal levels 3 min after addition of EGF. Concomitantly, Dab1 phosphorylation was induced but decreased to about 60% after 20 min (Figure 2A, Panel 1 and Figure 2B). To test whether this effect is specific for HEK293 cells, the experiment was repeated using NIH3T3 fibroblasts. As demonstrated in Figure 2C,D, the same pathway is functional in these cells and follows a similar kinetic as in HEK293 cells. Transfected fibroblasts in general express much less of exogenous proteins, proving that the effect seen in HEK293 cells is not due to overexpression of the components. Following this, we evaluated the presence of this EGF/EGFR/Dab1 pathway in primary neurons, cells which express all 3 components in a physiological context [27]. To calibrate this system we treated the cells in parallel with Reelin (Reelin-conditioned medium) and observed the well-established effect. Addition of Reelin led to a robust increase (4 fold) of Dab1 phosphorylation (Figure 2E, Lanes 1 and 2). This effect is mediated by ApoER2 and VLDL receptor and defines the canonical Reelin pathway [3]. As a control, we also tested for phosphorylation of EGFR upon Reelin stimulation. As seen in panel 3, Reelin did not increase activation of EGFR. Addition of EGF, however (Figure 2E, Lane 4, Panel 1), did not increase Dab1 phosphorylation in primary neurons. Phosphorylation of EGFR, however, was clearly induced and led to degradation of EGFR (Figure 2E, Lane 4, Panels 3 and 4).

Before exploring the details of the weakness of this effect in primary neurons, we evaluated how Dab1 becomes phosphorylated via the EGF/EGFR axis. Again, HEK293 cells and NIH3T3 fibroblasts were transiently transfected with plasmids expressing Dab1 and EGFR. Prior to stimulation with EGF, cells were treated either with Gefitinib or Neratinib and phosphorylation of EGFR and Dab1 was evaluated by Western blotting as described in Figure 3. Both compounds bind to EGFR targeting the ATP-binding pocket of the receptor inhibiting auto-phosphorylation of the receptor and consequently propagation of the signal [28]. This effect is clearly visible in both experiments (Figure 3A,B; Lanes 3 and 4, Panel 3). In addition Dab1 phosphorylation is also abolished (Figure 3A,B; Lanes 3 and 4, Panel 1) suggesting that EGFR activation is necessary for Dab1 activation. To further prove that Dab1 phosphorylation depends on the kinase activity of EGFR, we expressed Dab1 together with a mutated version of EGFR which lacks the kinase activity. In cells expressing the wild type form of EGFR (Figure 3C, Lanes 1 and 2) addition of EGF led to the activation of EGFR and increased Dab1 phosphorylation (Lane 2). In cells expressing the mutant form of EGFR (kinase-dead—KD, Lanes 3 and 4) addition of EGF neither caused EGFR activation nor Dab1 phosphorylation (Lane 4). These results suggest that the mechanism, by which Dab1 is phosphorylated by EGFR, is different from that mediated by ApoER2 or VLDLR. In the latter case, Dab1 is phosphorylated by Src family kinases and not by the receptors themselves. ApoER2 and VLDLR have no kinase domain and phosphorylation is mediated by receptor-mediated clustering of Dab1 [10]. EGFR becomes auto-phosphorylated and transmits the signal by either recruiting adapter proteins or directly activating downstream molecules by phosphorylation [29].

To further strengthen the hypothesis that EGF-mediated Dab1 phosphorylation is independent from the canonical Reelin pathway we tested whether EGF binds to ApoER2 or VLDLR and thus, would induce the canonical Reelin pathway by clustering the Reelin receptors. All three receptors (ApoER2, VLDLR, and EGFR) were expressed in HEK293 cells as mCherry fusion proteins and the cells were incubated with tagged EGF (Dylight488). Expression of the receptors (red channel) (Figure 4A,E,I), binding of EGF to the cells (green channel) (Figure 4B,F,J,N) and DAPI staining (Figure 4C,G,K,O,S) was visualized by confocal microscopy. As demonstrated in Figure 4B,D,F,H, ApoER2 and VLDLR do not bind EGF. As positive control, expression of EGFR produced a strong signal evoked by bound EGF-Dylight488 on the cell surface which co-localizes with EGFR (L). Panels M–P represent a control experiment where cells express soluble mCherry and EGF was added (N) as for B,F, and J. Panels Q–T represent another control experiment where cells express EGFR-mCherry without addition of EGF-Dylight488. Both control experiments do not produce significant background signals in the green channel used to detect binding of EGF.

The observation that the mechanism of EGF/EGFR-mediated phosphorylation of Dab1 is different from that of the canonical Reelin pathway prompted the idea that the very weak phosphorylation of Dab1 induced by EGF in primary neurons (see Figure 2E) might be caused by a competition of both pathways in these cells. To test this idea EGF/EGFR-induced phosphorylation of Dab1 in HEK293 cells was re-evaluated in the presence of ApoER2. EGF stimulation of the cells in the presence of ApoER2 led to a dramatic decrease of Dab1 phosphorylation without changing the phosphorylation level of EGFR (Figure 5A, compare Lanes 2 and 4). Statistical analysis of this effect revealed that the decrease is significant (Figure 5B). This result suggests that ApoER2 might compete for the binding of Dab1, reducing the amount of the adapter bound to EGFR and as a consequence reduces Dab1 phosphorylation by the EGF/EGFR pathway as seen in primary neurons from wild-type (wt)-mice (Figure 2E).

Having established that Dab1 is a novel adapter protein for EGFR and can be phosphorylated by EGF we tested whether Dab1 is involved in cell proliferation. HEK293 cells, which were used for this experiment, are of neuronal origin [30] and express small amounts of Dab1 [10]. Thus, we created a HEK293 Dab1 knock out cell line using CRISPR/Cas9. As demonstrated in Figure 6A (Lane 4) no expression of Dab1 could be detected in these cells (compared to wt cells; Lane 1). Control experiments omitting reverse transcriptase (Lane 2), without input mRNA (Lane 3), or without DNA (Lane 6) were performed in parallel to verify the absence of the band derived from Dab1-mRNA in the k.o. cells. To evaluate the absence of Dab1 in respect to cell proliferation, k.o. cells and wt cells were grown in parallel in a medium supplemented with fetal calf serum (FCS) and cell proliferation was measured using two different methods. First, an EdU-based assay directly measuring DNA synthesis was applied [31]. Here a fluorescent dye (Alexa Fluor 488) is incorporated into newly synthesized DNA which can be evaluated under the microscope. As demonstrated in Figure 6B, significantly less dividing cells are present when Dab1 is absent. The reduced proliferation rate could be re-established by expression of Dab1 (Figure 6A; Lane 5; Figure 6B). To further verify this result a metabolic assay (WST-1) was used [32]. This assay assesses proliferation and survival rates, and also resulted in a significant reduction of metabolic activity reflecting the number of viable cells when they lack Dab1 (Figure 6C). In this case, re-expression of Dab1 in either wt cells or k.o. cells led to a significant increase of number of cells, an effect which was not seen using the EdU-based assay. Such discrepancies are known and well described and are caused by an over-estimation of cell proliferation produced by metabolic assays [33]. Taken together, these experiments demonstrate that Dab1 is part of a signalling pathway which is involved in basic cell proliferation and/or cell survival.

To test whether this pathway is functional in vivo we turned to the small intestine for following reasons: enterocytes express Dab1 throughout the intestinal mucosa [22] and its expression is strongly induced in colorectal cancer [24]; furthermore, Paneth cells, which constitute the niche for intestinal stem cells in intestinal crypts, secrete Wnt3, TGFα, the Notch-ligand D114, and EGF [34]. Primary intestinal cells were prepared from the small intestine from adult mice and tested for the expression of Dab1, EGFR, VLDLR, and ApoER2. As demonstrated in Figure 7A, these cells express Dab1 (Lane 1) and EGFR (Lane 3), but no VLDLR (Lane 5) nor ApoER2 (Lane 7). Dab1 however, is expressed as truncated isoform missing exons 7 and 8, but carrying an alternative exon 9 coding for an insertion of 33 amino acids [35] migrating at around 70 kDa in the gel system used here (compare Lanes 1 and 2). Lane 2 exhibits the presence of full length Dab1 in mouse embryonic brain. The truncated variant present in intestinal cells lacks tyrosines 198, 200, and 220, but still contains tyrosine 232. To test whether the Dab1 variant present in intestinal cells is phosphorylated, Dab1 was precipitated with a Dab1-specific antibody (Ab 54, Figure 7A, Lane 9) or an unspecific IgG (Lane 10) and the precipitate was probed with an anti-phosphotyrosine antibody (Ab PY99). To prove that the phosphorylated protein seen in Lane 9 is indeed Dab1 the blot was stripped and re-probed for Dab1 with an antibody against Dab1 (D4).

To test whether the truncated Dab1 isoform present in intestinal cells can be phosphorylated by EGF, we expressed the corresponding isoform from chicken (this isoform was assigned as Dab1 early (Dab1E) [36]) in HEK293 cells expressing recombinant EGFR and treated the cells with EGF as described above. Stimulation with EGF induced activation of EGFR and, as already demonstrated in Figure 2 for embryonic neurons, led to its degradation (Figure 7B, Lanes 1–4, Panels 3 and 4). To test for Dab1E phosphorylation, Dab1E was precipitated and its phosphorylation evaluated using a p-Tyr specific antibody. As seen in Lanes 3 and 4, Dab1E is indeed phosphorylated by EGF. To test whether the observed phosphorylation is on Tyr232, the cell extracts were directly tested by Western blotting using a Dab1-tyrosine 232 specific antibody (Figure 7B, Lanes 5–8). To verify that the band migrating at around 80 kDa (arrow in Figure 7B, panel 1) is indeed chicken Dab1E, extracts derived from untransfected cells were evaluated in parallel (Figure 7B, Lanes 5 and 6). The bands seen in Lanes 6 and 8 (100 kDa and below 70 kDa) must represent proteins expressed in HEK293 cells which become phosphorylated by EGF and are unspecifically detected by the antibody used. Phosphorylation of Dab1E by EGF is a very interesting observation, since it was published that this isoform cannot be phosphorylated by the Reelin/ApoER2/VLDLR pathway [37]. To prove this observation, full length Dab1 (555) and Dab1 early were expressed in HEK293 cells also expressing ApoER2. These cells were treated with Reelin and as demonstrated in Figure 7C, Dab1E is indeed not phosphorylated (Lane 4) by the Reelin/ApoER2 pathway.

Taken together, these results indicate that in primary intestinal epithelial cells a Reelin-independent pathway is operating resulting in the phosphorylation of a truncated Dab1 isoform possibly mediated by EGF and EGFR.

## 3. Discussion

Since Dab1 is involved in signalling events leading to cell proliferation independent of the canonical Reelin pathway, we turned our attention to receptor tyrosine kinases (RTKs). This family of receptors harbor a NPxY motif in their intracellular domain (for review see [38]). Signaling via RTKs is initiated on binding soluble homo- or hetero-dimeric ligands leading to dimerization or oligomerization of the receptors and auto-phosphorylation of their ICDs. The signal is propagated by binding of different adapter molecules containing an SH2 or phosphotyrosine-binding domain (PTB) binding domain. The canonical binding site for PTB-domain-containing proteins is the NPxY motif [39]. Three different classes of PTB-containing proteins exist; one of them contains the so-called “Dab-like” PTB domain. Binding of this domain is not dependent on phosphorylation of the NPxY motif and is mediated by hydrophobic contacts and hydrogen bonds between the PTB domain of the adapter and the NPxY motif of the receptor [40]. Here we show that Dab1 indeed binds to EGFR and can be phosphorylated by the addition of EGF. This novel pathway is independent upon Reelin and the presence of ApoER2 and/or VLDLR and thus, constitutes a hitherto unknown signalling pathway. The mode of phosphorylation of Dab1 in the EGF/EGFR pathway is different from that established for the canonical Reelin pathway [3]. In the Reelin pathway phosphorylation of Dab1 is induced by receptor clustering [10,11], rendering the bound Dab1 to a substrate for Src family kinases [8,9]. In the EGF/EGFR pathway, Dab1 phosphorylation is dependent on the kinase activity of EGFR since it can be blocked by inhibitors of this activity such as Gefitinib and Neratinib.

As demonstrated in Figure 1C, co-precipitation of Dab1 with EGFR from brain extracts was much weaker than from extracts derived from HEK293 cell expressing both components (Figure 1B). In addition, phosphorylation of Dab1 by the EGF/EGFR pathway in embryonic neurons is not induced (Figure 2E). We have evaluated these results by co-expressing ApoER2 in the HEK239 cell system leading to a significant decrease of Dab1 phosphorylation induced by EGF. These results suggest that ApoER2 competes for Dab1 binding reducing the amount bound to EGFR and inhibiting phosphorylation of Dab1 by EGF/EGFR. This idea is supported by the fact that the ideal binding site for Dab1 is ΦXNPXY (where Φ is either F or Y) [7]. EGFR contains 3 potential PTB binding sites in its intracellular domain. None of them, however, contains F or Y in this particular position. Thus, the affinity of Dab1 to ApoER2 might be significantly higher than to EGFR. This situation might create a hierarchy of signaling events favoring the Reelin signalling pathway in the brain.

To establish a potential role of this pathway in cell proliferation, we created Dab1 k.o. HEK293 cells. These cells are indeed characterized by a reduced proliferation rate which can be rescued by re-expressing Dab1. In the search for a suitable physiological cell system to prove that the EGF/EGFR/Dab1 pathway is functional in vivo we turned our attention to intestinal cells for the following reasons: the lining of the intestine is renewed at a very high rate [41], the stem cell niche which is responsible for this high proliferation rate contains EGF [34], rat intestinal cells express Dab1 [22], and most importantly Dab1 is overexpressed in intestinal cancer and promotes colorectal cancer cell invasion and metastasis [24]. To this end, we isolated epithelial cells from mouse gut and tested them for the expression of Dab1, EGFR, VLDLR, and ApoER2. It is known that EGF activates proliferation and blocks apoptosis of mouse intestinal progenitor cells in culture [42]. Intestinal cells express a truncated isoform of Dab1 as demonstrated in Figure 7A, in agreement with previously published results [35]. This isoform lacks exons 7 and 8 thus missing tyrosines 198, 200, and 220, all of them phosphorylated by the canonical Reelin pathway. The remaining tyrosines 185 and 232 are not phosphorylated by the Reelin/ApoER2 axis in the absence of tyrosines 198, 200, and 220 [37]. This fact is important for evaluating the observation that the truncated Dab1 isoform present in intestinal cells is phosphorylated. These cells express EGFR, but neither VLDLR nor ApoER2 as demonstrated in Figure 7A. This result is not in agreement with a previously published publication claiming that ApoER2 and VLDLR are expressed in these cells [22]. Nevertheless, if these receptors and Reelin are indeed present, this pathway is not expected to phosphorylate truncated Dab1 as discussed above. In addition to the fact, that EGFR is activated in intestinal cells [42] and EGF is able to phosphorylate truncated Dab1 in transformed HEK293 cells expressing EGFR suggests that Dab1 might be phosphorylated by activated EGFR in this system.

As mentioned above, one of the genes induced by Notch signalling in colorectal cancer is Dab1 driving cancer cell proliferation and invasion by activation of a Dab1-Abl-RhoGEF protein Trio pathway [24]. It is not clear yet, which variant of Dab1 is over-expressed in this situation but exclusion of exons 7 and 8 combines exon 6 with exon 9 re-establishing an Abl/Crk binding motif at tyrosine 185 and maintaining the Abl/Crk motif at tyrosine 232. Both Abl/Crk recognition motifs are present in full length Dab1 which harbors two SFK binding motifs in addition [43]. Thus, both variants might be able to take part in the proposed Dab1-Abl-RhoGEF protein Trio pathway [24]. Dab1 overexpressed in T-lymphoblastic leukemia cells, however, misses exons 7 and 8 [25] and might drive proliferation of these cells in a similar way.

Together with the observation that the scrambler mutation (Dab1-/-) reduces the number of Paneth cells in the intestinal crypts [44] our results suggest that besides the well-established Wnt-pathway which is the major player in the proliferation of intestinal stem and precursor cells, the EGFR/Dab1 pathway might be involved in the extraordinary renewal rate of these cells under physiological conditions. Down-regulation of this pathway might cause diminished barrier function (for review see [45]), whereas over-activation might lead to cancer formation.

The idea that the newly discovered EGFR/Dab1 pathway might play a more general role in cell proliferation is supported by previous observations that truncated Dab1 is involved in proliferation and maintenance of the retinal progenitor pool in the developing eye [37,46] and in the rapid expansion of granulosa cells in the developing follicle in egg laying species [47].

## 4. Materials and Methods

### 4.1. Animals

Wt mice on a C57BL6/J background [5] were housed in strict accordance with prevailing guidelines for animal care and welfare. Animal handling and sacrificing were approved by the Austrian Federal Ministry of Science and Research (permit number, BMWFW-66.006/0012-WF/II/3b/2014; date of issue 22 May 2015).

### 4.2. Primary Murine Embryonic Neuron Culture

To obtain primary neuronal cells, male and female mice were mated. If a vaginal plug was observed the next morning, it was designated as embryonic day (E0.5). Embryos were collected at day E15.5 from a pregnant dam and brains were obtained. Meninges were discarded and forebrains were collected in Hibernate^®^-E (Thermo Scientific, Waltham, MA, USA) supplemented with 2% B-27^®^ Serum-Free Supplement (Thermo Scientific) and 0.5 mM GlutaMAX™-I (Thermo Scientific) at 4 °C. Next, brains were washed two times with Hanks’ Balanced Salt Solution (HBSS, PAN-BioTech, Aidenbach, Germany). Afterwards, they were thoroughly resuspended in Neurobasal medium supplemented with 2% B-27^®^ Serum-Free Supplement (Thermo Scientific) and 0.5 mM GlutaMAX™-I (Thermo Scientific) and seeded on poly L-ornithine coated plates. Every three-four days half of the medium was aspirated and replaced with fresh Neurobasal medium. At days in vitro 7 (DIV7) neurons were treated with RCM/MCM or EGF as described below.

### 4.3. Isolation of Mucosal Epithelial Cells

Isolation of mucosal epithelial cells from small intestine from adult wt mouse was performed according to [48]. An adult male mouse was sacrificed and the skin of the abdominal region was washed with 70% ethanol and cut open. Small intestine was resected and washed in ice-cold HBSS. Fat and adherent connective tissues were removed and intestinal tissue was cut longitudinally and washed three times with ice-cold HBSS. Then, it was washed five times in 10 mL of ice-cold HBSS in a 50 mL falcon. Subsequently, it was put in a 15 cm dish containing 10 mL HBSS, completely unfolded, slightly stretched, and the intestinal mucosa was scraped off very gently with a blunt-edge of a metal blade. Intestinal cells were transferred to a falcon tube, centrifuged at 300× *g* for 5 min at 4 °C and supernatant was aspirated. Cells were thoroughly re-suspended in 5 mL of ice-cold HBSS and spun down at 1400× *g* for 5 min at 4 °C. Supernatant was aspirated and the pellet was lysed in NP-40 lysis buffer supplemented with complete^™^ EDTA-free Protease Inhibitor Cocktail (Roche, Basel, Switzerland), PhosSTOP (Roche), and 1 mM EDTA. Lysates were flash frozen in liquid nitrogen and stored at −80 °C.

### 4.4. Cloning

Plasmids encoding pClneo_EGFR and pmCherry_EGFR were constructed by PCR amplification of the cDNA for EGFR from pHom1_EGFR_mGFP [49]. For construction of pClneo_EGFR the following primers were used: 5′- atatgtcgacatgcgaccctccgggacg -3′ and 5′- atatgcggccgctcatgctccaataaattcactgctttg -3′, which introduced flanking restriction sites *SalI* and *NotI* (underlined). The EGFR PCR product was inserted into the corresponding sites of pClneo to produce pClneo_EGFR. For construction of pmCherry_EGFR the following primers were used: 5′- atatctcgagatgcgaccctccgggacg-3′ and 5′- atataagctttgctccaataaattcactgctttgtgg -3′, which introduced flanking restriction sites XhoI and HindIII (underlined). The EGFR PCR product was inserted into the corresponding sites of pmCherry_N1 (Clontech) to produce pmCherry_EGFR. To generate pmCherry-N1_VLDLR cDNA for mmVLDLR (murine VLDLR lacking the O-linked sugar domain, which is the predominant splice form in murine brain [50]), was amplified and inserted into pmCherry-N1 which was digested with *SalI* and *HindIII*. pClneo_VLDLR and pClneo_ApoER2 were constructed as described in [50,51].

pmCherry-N1_ApoER2 and pmGFP were constructed as described in [10]. To construct pDab1_mGFP the Dmr domain from pHom1_Dab1_mGFP was removed by digestion with *XbaI* and *SpeI* and self-ligation. To generate pHom1_Dab1_mGFP mmDab1_555 cDNA was amplified by PCR using primers 5′- cctagagaattcatgtcaactgagacagaac -3′ and 5′- taagcatctagagctaccgtcttgtggac -3′ and was inserted into pHom1_mGFP digested with *EcoRI* and *XbaI*. pHom1_EGFR_mGFP was a kind gift from Paul M.P. van Bergen en Henegouwen (Utrecht University). pDSred_Dab1_555 (with codon stop before RFP ORF) was a kind gift of Brian W. Howell (Upstate Medical University). pCDNA3.1_chDab1E was a kind gift of Rosaline Godbout (University of Alberta). pBABE_puro_ K721A-EGFR and pBABE_puro_EGFR_WT were a kind gift of Maria Sibilia (Medical University of Vienna). Dab1 CRISPR/Cas9 KO Plasmid (h), sc-402167, was purchased from Santa Cruz Biotechnology, Inc (Dallas, TX, USA). Fidelity of all constructs was tested by sequencing.

### 4.5. Creation of HEK293 Dab1 k.o. Cell Line by CRISPR/Cas9

HEK293 cells were seeded on 12-well plate and after 24 h transfected with Dab1 CRISPR/Cas9 KO Plasmid (sc-402167, Santa Cruz Biotechnology) using UltraCruz Transfection Reagent (sc-395739, Santa Cruz Biotechnology) according to the manufacture’s protocol. At 48 h after transfection, cells were detached and sorted by FACS in a 96-well plate coated with 50 μg/mL of collagen containing 100 μL of filtered HEK293 conditioned DMEM + 10% FCS (Sigma, St. Louis, MO, USA) medium per well. Cells were incubated at 37 °C and expanded to 10 cm dishes. RNA was extracted from the cells by using Monarch Total RNA Miniprep Kit (#T2010G, NEB). Thermo Fisher RevertAid H Minus First Strand cDNA Synthesis Kit was applied to synthesize cDNA from the purified RNA. PCR was performed using the forward primers 5′- TAAGCAGAATTCCCATGTCAACTGAGACAGAAC -3′, 5′- CTAGCTACCGGCCTGTG -3′ for Dab1 as well as a forward and reverse primer for GAPDH included in the Thermo Fisher RevertAid H Minus First Strand cDNA Synthesis Kit. The resulting PCR products were analyzed using 1% agarose gel.

### 4.6. Cell Lines and Transfection

HEK293 (Human embryonic kidney cells 293, ATCC), HEK293T (Human embryonic kidney cells 293, ATCC CRL-3216) and NIH3T3 (Murine Fibroblasts, ATCC, Manassas, VA, USA) were cultivated in Dulbecco’s modified Eagle’s medium (DMEM; Gibco) supplemented with 10% fetal calf serum (Sigma) at 37 °C and 5% CO_2_. After 24 h cultivation, HEK293 cells were transfected using PEI (polyethylenimine) and the indicated constructs according to [52]. HEK293T cells were transfected using the jetPRIME^®^ transfection kit from Polyplus transfection^®^ accordingly to the manufacturer’s protocol. NIH3T3 cells were transfected using Ultracruz^®^ transfection kit from Santa Cruz Biotechnology (sc-395739) accordingly to the manufacturer’s protocol. Cells were incubated for 24 h, followed by treatments, cell lysis or fixation.

### 4.7. EGF Treatment

Cells were washed and incubated in starvation medium consisting of 4 mM Glutamine in Hank’s Balanced Salt solution (HBSS) for 30 min at 37 °C + 5% CO_2_. Afterwards, the dishes were supplemented with 10 ng/mL of human EGF (PHG0315, Thermo Scientific, Waltham, MA, USA) or 20 ng/mL of murine EGF (130-094-036, Miltenyi Biotec, Bergisch Gladbach, Germany) and were incubated for 3, 10, or 20 min at 37 °C + 5% CO_2_. One dish was left untreated and served as negative control. In case of the inhibitor treatment, 1 µM Neratinib (MCE, Cat. No.: HY-32721) or 1µM Gefitinib/Iressa (MCE, Cat. No.: HY-50895) was added 2 h prior to starvation. During starvation and EGF treatment, again 1 µM of the inhibitors was added and the cells incubated at 37 °C + 5% CO_2_. Then, EGF treatment was conducted as described above.

### 4.8. Preparation of Cell Extracts, SDS-PAGE, Western Blotting, Immunoprecipitation

Cell extracts were prepared in NP-40 lysis buffer (150 mM sodium chloride, 1.0% Nonidet P-40, 10% glycerol, 20 mM Tris, pH 7.4) supplemented with complete^™^ EDTA-free Protease Inhibitor Cocktail (Roche), 0.05 mM NaF, 1 mM Na_3_VO_4_ and 1 mM EDTA and were used directly for SDS-PAGE or for immunoprecipitation. Extracts containing 400–500 µg of protein were incubated for 1 h at 4 °C with anti-Dab1 (Ab 54), anti-EGFR, anti-ApoER2 (Ab 186) or an unspecific anti-mouse/anti-rabbit IgG antibody and then for 1 h with Protein A Sepharose 4B (Invitrogen, Carlsbad, CA, USA). Beads were collected by centrifugation at 500× *g* for 1 min and washed three times using NP-40 lysis buffer supplemented with Protease Inhibitor Cocktail, 1 mM EDTA, 0.05 mM NaF and 1 mM Na_3_VO_4_. Bound proteins were eluted with 60 µL of 4× SDS protein sample buffer and the samples were boiled for 3 min at 96 °C. The samples were centrifuged for 5 min at 17.000× *g* at room temperature and the supernatant was collected. Proteins were separated by reducing SDS-PAGE and transferred onto Amersham™ Protran nitrocellulose membrane (GE Healthcare, Chicago, IL, USA) by wet blotting. Membranes were blocked in PBS or TBS containing 0.1% Tween-20 and 5% bovine serum albumin and incubated with primary antibody over night at 4 °C. After washing membranes were incubated with horseradish peroxidase (HRP)-conjugated secondary antibodies (Jackson ImmunoResearch Laboratories, West Grove, PA, USA). For detection, enhanced chemiluminescence solutions, NOVA 2.0, Supernova (Cyanagen, Bologna, Italy), were used. Blots were stripped by incubating for 30 min in a stripping buffer consisting of 25 mM Glycine (pH = 2) with 2% SDS. The blots were washed 3 times in PBS or TBS + 0.1% Tween-20, blocked in 5% BSA, PBS/TBS+ 0,1% Tween and incubated with the primary antibody over night at 4 °C. After washing membranes were incubated with HRP-conjugated secondary antibodies (Jackson ImmunoResearch Laboratories, West Grove, PA, USA).

### 4.9. EGF Labelling

HEK293 wt cells were seeded on a 24 well plate with 12 mm sterile glass coverslips added and coated with 50 µg/mL collagen I (A10483-01, Gibco). After 24 h, when cells were 50–70% confluent, they were transfected with VLDLR_mCherry, ApoER2_mCherry, EGFR_mCherry, or pmCherry by using PEI reagent. After 24 h cells were incubated for 30 min with imaging medium (Hank’s Balanced Salt solution, 4 mM Glutamine) and human EGF (PHG0315, Thermo Scientific) which was labelled with Dylight 488 (Thermo Scientific 53025) according to the protocol provided by the supplier. A total of 100 µL of EGF with a concentration of 1 mg/mL was used for labelling and immediately added to the cells and incubated for exactly 1 min and 30 s at 37 °C and 5% CO_2_. Cells were washed with PBS and fixed with 4% formaldehyde for 15 min at RT in darkness. Cells were washed three times with PBS and stained with DAPI (4′,6-diamidino-2-phenylindole, 5 µg/mL) for 5 min. After another washing step with PBS they were quenched with glycine (100 mM) for 15 min. Cells were washed twice with H_2_O and mounted with 3 µL DAKO fluorescent mounting medium (DAKO S3023, Lot 10067534) and sealed with nail polish. Pictures were acquired with a Laser Scanning Confocal microscope (LSM 700, ZEISS) using ZEN software.

### 4.10. EdU Staining

HEK293 wt and Dab1 k.o. cells were seeded at a density of 3.5 × 10^5^ cells/mL in a 12 well plate. When they had reached 50–70% confluency, they were transfected via PEI with pClneo or Dab1_555. After 24 h half of the medium was replaced by fresh DMEM, containing 10% FCS and 20 µM EdU (5-ethynyl-2’-deoxyuridine, final concentration: 10 µM/stock: 10 mM, Invitrogen, C10337). Cells were incubated at 37 °C for 2 h and fixed for 15 min at RT with 3.7% FA. They were washed 2 times with 3% BSA/PBS and permeabilized with 0.5% Triton X-100 in PBS for 20 min. The Click-iT reaction cocktail was mixed 10 min before it was used according to the provided protocol by Invitrogen. Cells were washed 2× and incubated with the reaction cocktail for 30 min, protected from light. Another washing step was applied, first with 3% BSA/PBS and then with PBS only. Cells were stained with 5 µg/mL DAPI, and incubated for 30 min. After washing twice with H_2_O they were mounted with 5 µL of mounting medium and sealed with nail polish. Pictures were acquired with Laser Scanning Confocal microscope (LSM 700, ZEISS, Jena, Germany) using ZEN software. Images were analyzed via Fiji/ImageJ. EdU and DAPI channels were split, the threshold was adjusted and pictures were processed to binary. To separate nuclei that were too close to each other watershed option was applied. Particles with a size bigger than 50 µm^2^ were counted.

### 4.11. Preparation of Reelin Conditioned Medium

Reelin-expressing HEK293 cells were cultivated and used for production of Reelin conditioned medium (RCM) as described before [53]. Briefly, HEK293 cells stably carrying the full-length mouse Reelin expression construct pCrl (a kind gift of Tom Curran, Perelman School of Medicine at the University of Pennsylvania, Philadelphia, PA, USA) were cultivated in DMEM + GlutaMax (Thermo Scientific) supplemented with 10% fetal calf serum (Sigma) and 0.5 mg/mL G418 (Roth) at 37 °C and 5% CO_2_. When the cells reached 70% confluency the culture medium was replaced by serum-free medium (OptiMEM, Gibco). After two more days the conditioned medium was collected, sterile filtered and used for experiments.

### 4.12. WST-1 Cell Proliferation/Survival Assay

HEK293 wt and HEK293 Dab1 k.o. cells were seeded with the same density on a 96 well plate coated with 50 µg/mL collagen I (A10483-01, Gibco). When 50–60% confluency was reached cells were transfected with Dab1_555 or pClneo using PEI transfection reagent. After 24 h they were treated according to the WST-Assay Protocol (Roche, 11644807001, Version August 2018) and 10 µL of WST-1 Reagent was added (100 nM stock) to each well. Absorbance was measured with a Victor^TM^ Nivo Plate reader after 30, 60, 120, and 180 min. For quantification; 600/10 nm absorbance and blank values were subtracted from 480/30 nm absorbance.

### 4.13. Statistical Analysis

All statistical tests were performed using GraphPad Prism version 6. Prior to analysis, normal distribution of the data was checked by the Shapiro–Wilk normality test. To compare two unpaired sample groups, two-tailed Student t-test was performed. To compare more than two sample groups one way ANOVA multiple comparison test was used. Results were considered significant when * *p* ≤ 0.05, ** *p* ≤ 0.01, *** *p* ≤ 0.001 and **** *p* ≤ 0.0001. Graphs are presented as mean ± standard deviation. ImageLab software (Bio-Rad) was used to quantify the signal intensity of Western blot bands.

### 4.14. Antibodies

Antibodies used in this study are summarized in Table 1.

## Figures and Tables

**Figure 1 ijms-22-01745-f001:**
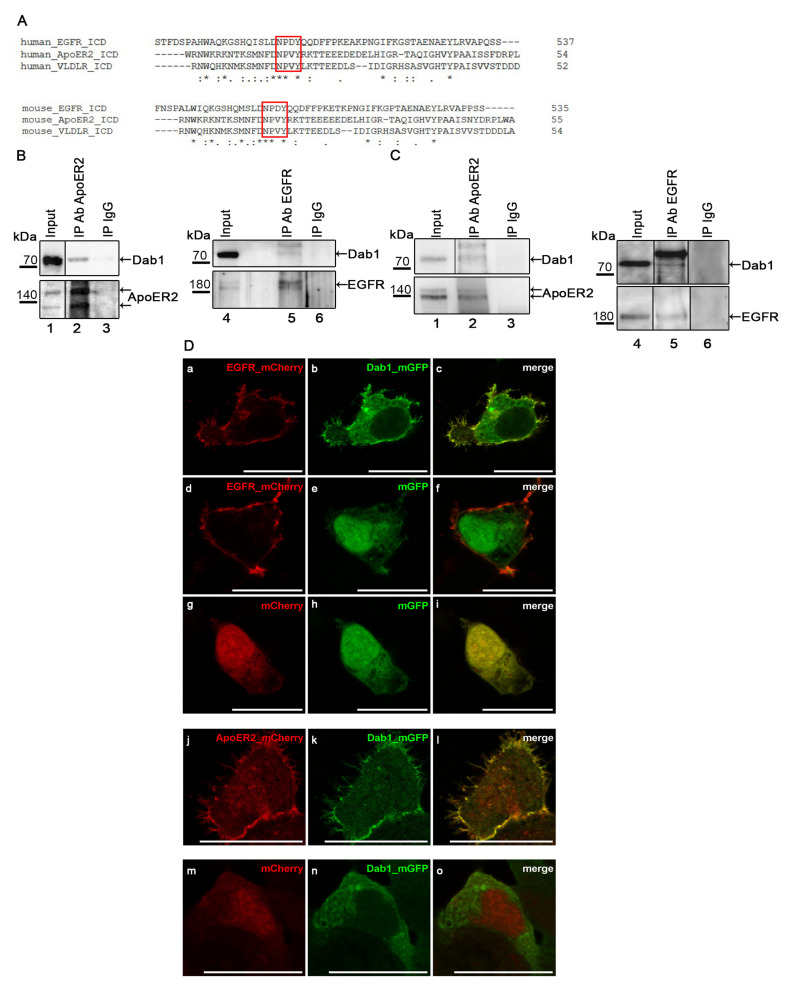
Disabled 1 (Dab1) interacts with epidermal growth factor receptor (EGFR). (**A**) Sequence alignment of human/murine apolipoprotein E receptor 2 (ApoER2), very low density lipoprotein receptor (VLDLR) and EGFR intracellular domains was generated with Clustal Omega. NPxY motifs present in the intracellular domains of the receptors are marked by red boxes. “*” residues are identical in all sequences in the alignment, “:” conserved substitutions, “.” semi-conserved substitutions. (**B**) Western blot analysis of protein extracts from HEK293 cells expressing ApoER2 and Dab1 (Lane 1, input) or EGFR and Dab1 (Lane 4, input) which were subjected to immunoprecipitations using an antibody specific to ApoER2 (Lane 2) or EGFR (Lane 5) or an unrelated rabbit IgG antibody (Lane 3) or mouse IgG antibody (Lane 6). In upper panels Dab1 levels were detected by using a Dab1 specific antibody (Ab D4). The blots were stripped and re-probed with an ApoER2 or EGFR specific antibody (lower panels) as a pulldown control. (**C**) Murine embryonic neurons (E15.5) were lysed and the protein extract was subjected to immunoprecipitation using an antibody specific to ApoER2 (Lane 2) or EGFR (Lane 5) or unrelated rabbit (Lane 3) or mouse IgG (Lane 6) as a control. The extracts and precipitates were analyzed by western blotting. The vertical line indicates that the blot was spliced. (**D**) HEK293 cells expressing fluorescently tagged variants of ApoER2 (**j**), EGFR (**a**,**d**) and Dab1 (**b**,**k**,**n**) or fluorophores alone; mGFP (**e**,**h**) or mCherry (**g**,**m**) were imaged by confocal fluorescence microscopy. Merged images are presented in (**c**,**f**,**i**,**l**,**o**). Scale bar represents 20 µm.

**Figure 2 ijms-22-01745-f002:**
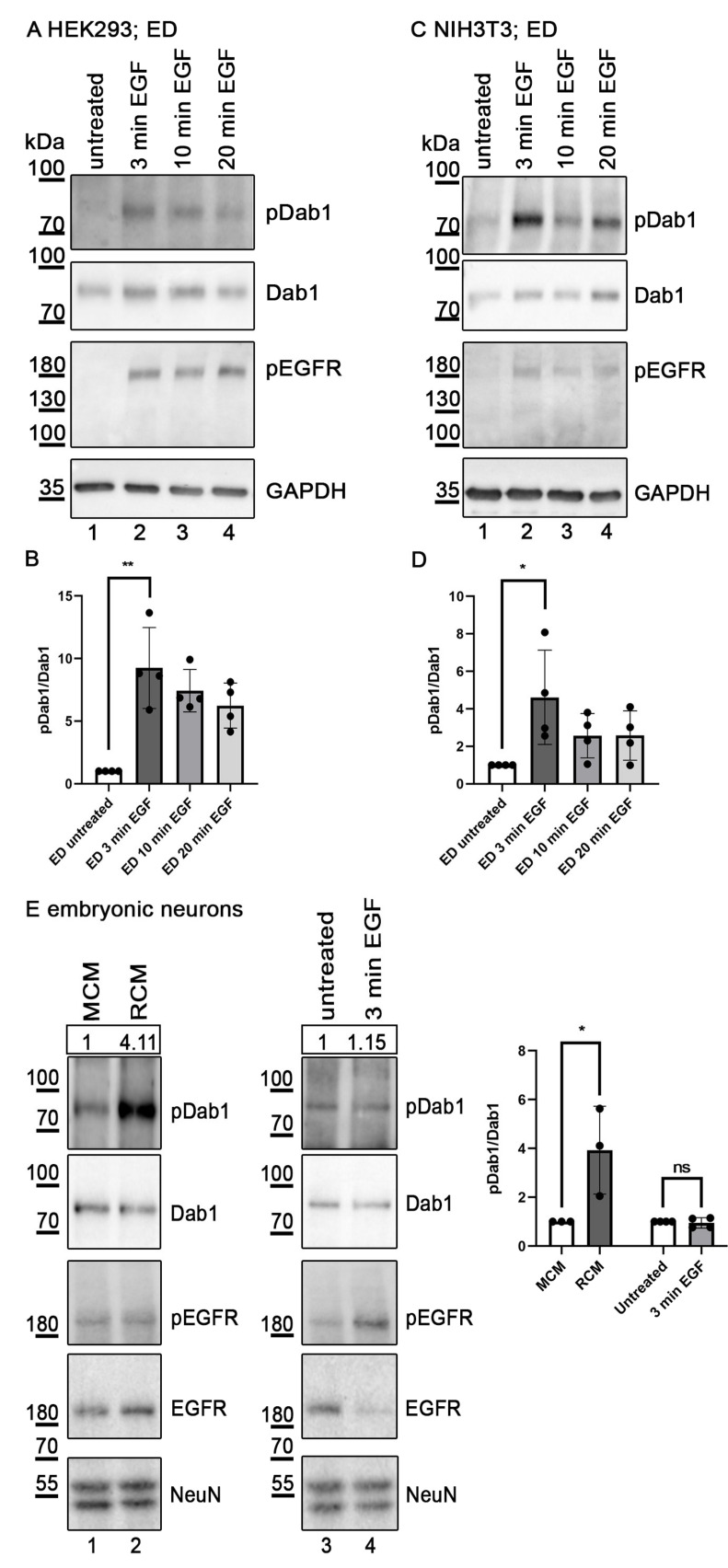
EGF induces Dab1 phosphorylation. Western blot analysis of protein extracts from HEK293 (**A**) and NIH3T3 (**C**) cells expressing EGFR and Dab1. Cells were either left untreated (Lane 1) or treated with human EGF (10 ng/mL) for 3 (Lane 2), 10 (Lane 3), or 20 min (Lane 4). Immunoprecipitation using a Dab1 specific antibody was performed (Ab 54). Dab1 phosphorylation levels (Panel A1/C1) were detected using an antibody against phosphorylated tyrosine residues (Ab PY99). The blots were stripped and re-probed for Dab1 using Ab D4 (panel A2/C2). Extracts were blotted with an antibody specific for phosphorylated EGFR (Tyr 1173) (panel A3/C3). As a loading control, GAPDH was used (Panel A4/C4). Anti-phosphotyrosine signal was normalized to the anti-Dab1 signal and relative levels of phosphorylated Dab1 from 4 independent experiments in HEK293 (**B**) and NIH3T3 (**D**) are presented. The levels of Dab1 phosphorylation in the untreated controls were set to 1. Data were analyzed by an unpaired, two-tailed *t*-test. (**E**) Primary neuronal cultures (DIV7) were treated with Reelin conditioned medium (RCM, Lane 2), Mock conditioned medium (MCM, Lane 1) for 20 min or with 20 ng/mL murine EGF for 3 min (Lane 4) or left untreated (Lane 3). Cells were lysed and the protein extracts were subjected to immunoprecipitation using an antibody specific to Dab1 (Ab 54). Dab1 phosphorylation levels (Panel 1) were detected using an antibody against phosphorylated tyrosine residues (Ab PY99). The blot was stripped and re-probed for Dab1 using Ab D4 (Panel 2). Extracts were blotted with an antibody specific for phosphorylated EGFR (Tyr 1173) (Panel 3), total levels of EGFR (Panel 4). As a neuronal loading control, Ab NeuN was used (Panel 5). The extracts and precipitates were analyzed by western blotting. Bands were scanned with ChemiDoc Touch Imaging System (BioRad), the anti-phosphotyrosine signal was normalized to the anti-Dab1 band and relative levels of phosphorylated Dab1 from 3–4 independent experiments are presented. The levels of Dab1 phosphorylation in the untreated or mock controls were set to 1. Relative intensity of the presented blot is shown in a box above the first panel. Data were analyzed by way of an unpaired, two-tailed *t*-test, * *p* ≤ 0.05, ** *p* ≤ 0.01, ns, not significant; dotes, number of experiments. Error bars represent standard deviation.

**Figure 3 ijms-22-01745-f003:**
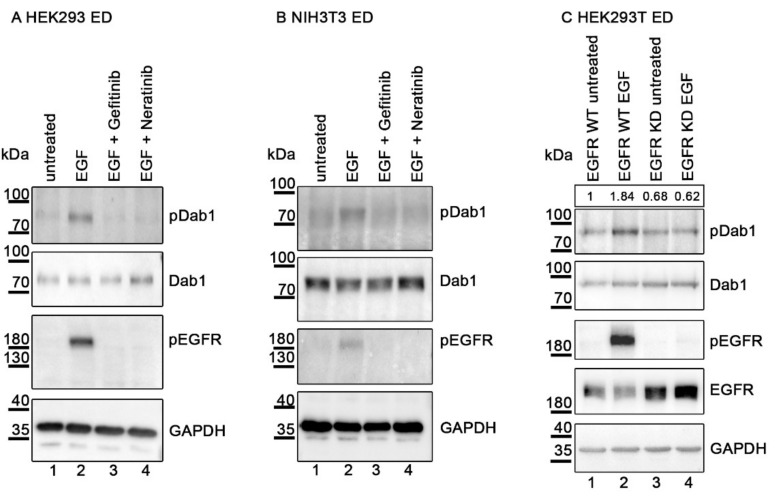
Dab1 phosphorylation depends on EGFR kinase activity. Western blot analysis of protein extracts from HEK293 (**A**) and NIH3T3 (**B**) cells expressing EGFR and Dab1. Inhibitors of EGFR phosphorylation, Gefitinib (Lane 3) and Neratinib (Lane 4) were added 2 h before EGF treatment (3 min at 37 °C). Lane 1 represents untreated negative controls, Lane 2 represents the EGF treated positive controls. Immunoprecipitation using a Dab1 specific antibody was performed (Ab 54). (**C**) Western blot analysis of protein extracts HEK293T expressing Dab1 and either a wild type (WT)-like EGFR or a kinase-dead (KD) EGFR. Lanes 1 and 3 represent the untreated negative controls and Lanes 2 and 4 represent EGF treatment. Immunoprecipitation using a Dab1 specific antibody was performed (Ab 54). Bands were scanned with ChemiDoc Touch Imaging System (BioRad, Hercules, CA, USA) and the anti-phosphotyrosine signal was normalized to the anti-Dab1 band. The level of Dab1 phosphorylation in the untreated WT EGFR (lane 1) control was set to 1. Relative intensity is shown in a box above the first panel. Dab1 phosphorylation levels (Panel 1) were detected using an antibody against phosphorylated tyrosine residues (Ab PY99). The blots were stripped and re-probed for Dab1 using Ab D4 (Panel 2). Extracts were blotted with an antibody specific for phosphorylated EGFR (Tyr 1173) (Panel 3) and EGFR (C, Panel 4). As a loading control GAPDH was used (Panel 4 in A and B or 5 in C).

**Figure 4 ijms-22-01745-f004:**
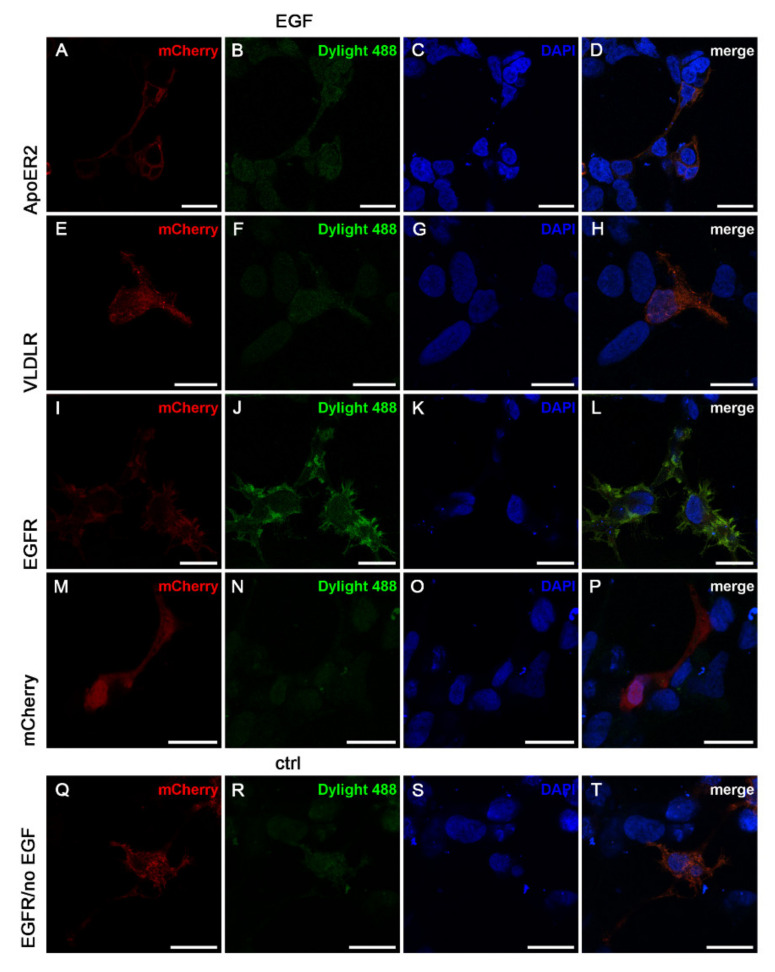
ApoER2 and VLDLR do not bind EGF. HEK293 cells expressing ApoER2_mCherry (**A**–**D**), VLDLR_mCherry (**E**–**H**), EGFR_mCherry (**I**–**L**) or mCherry alone (**M**–**P**) were incubated with EGF labelled with Dylight488 for 1 min at 37 °C (EGF). In the control condition (ctrl) no EGF was added to EGFR_mCherry expressing cells (**Q**–**T**). After incubation, cells were fixed, stained with DAPI and mounted. Scale bar represents 20 µm.

**Figure 5 ijms-22-01745-f005:**
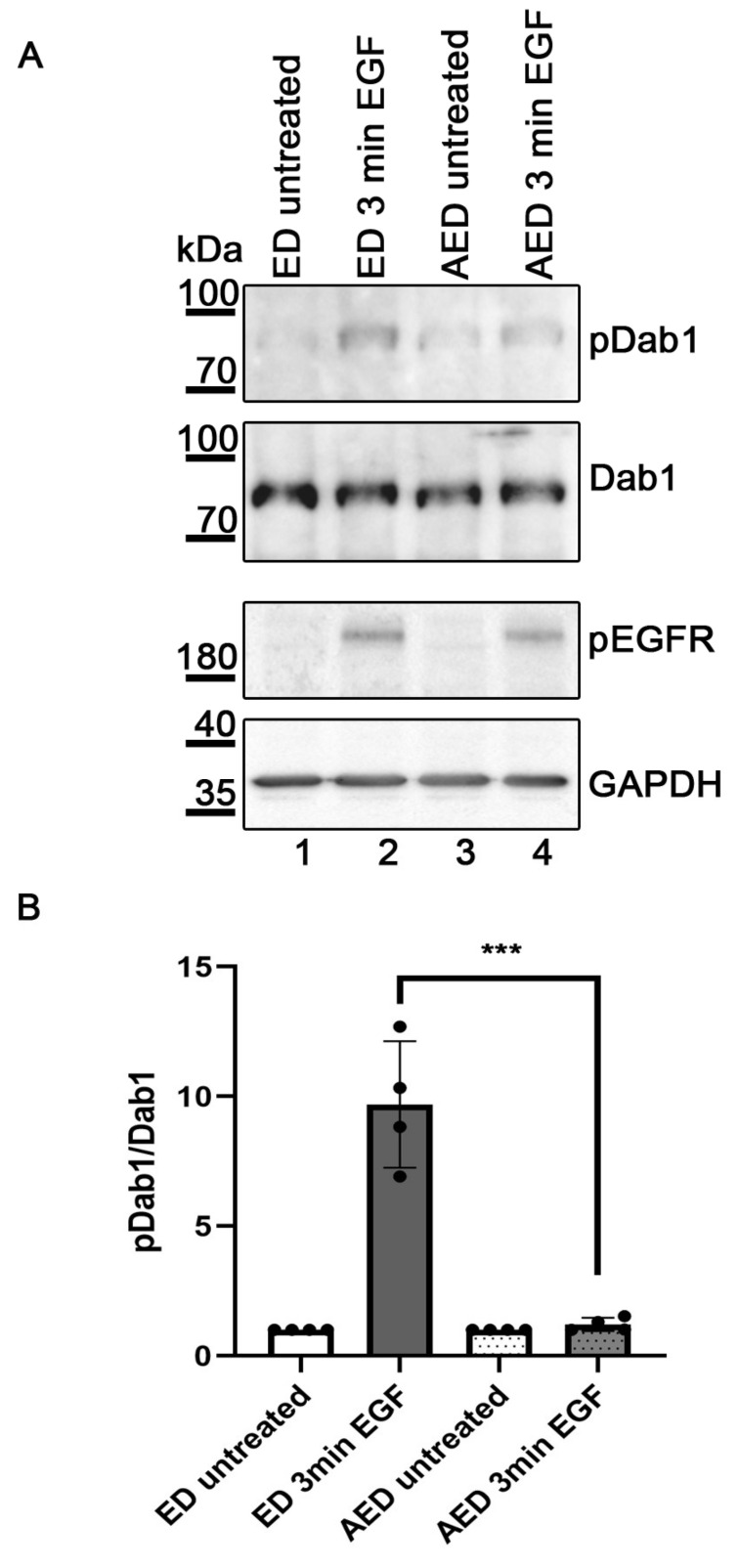
Presence of ApoER2 diminishes EGF-mediated Dab1 phosphorylation. Western blot analysis of HEK293 (**A**) cells expressing EGFR and Dab1 (Lanes 1 and 2, ED) or ApoER2, EGFR and Dab1 (Lanes 3 and 4, AED). Cells were either left untreated (Lanes 1 and 3) or treated with human EGF (10 ng/mL) for 3 min (Lanes 2 and 4). Immunoprecipitation using a Dab1 specific antibody was performed (Ab 54). Dab1 phosphorylation levels (Panel 1) were detected using an antibody against phosphorylated tyrosine residues (Ab PY99). The blots were stripped and re-probed for Dab1 using Ab D4 (Panel 2). Extracts were blotted with an antibody specific for EGFR phosphorylation (Tyr 1173) (Panel 3) and GAPDH was used as a loading control (Panel 4). (**B**) The anti-phosphotyrosine signal was normalized to the anti-Dab1 signal and relative levels of phosphorylated Dab1 from 4 independent experiments in HEK293 are presented. The level of Dab1 phosphorylation in the untreated control was set to 1. Data were analyzed by an unpaired, two-tailed *t*-test, *** *p* ≤ 0.001; dots, number of experiments. Error bars represent standard deviation.

**Figure 6 ijms-22-01745-f006:**
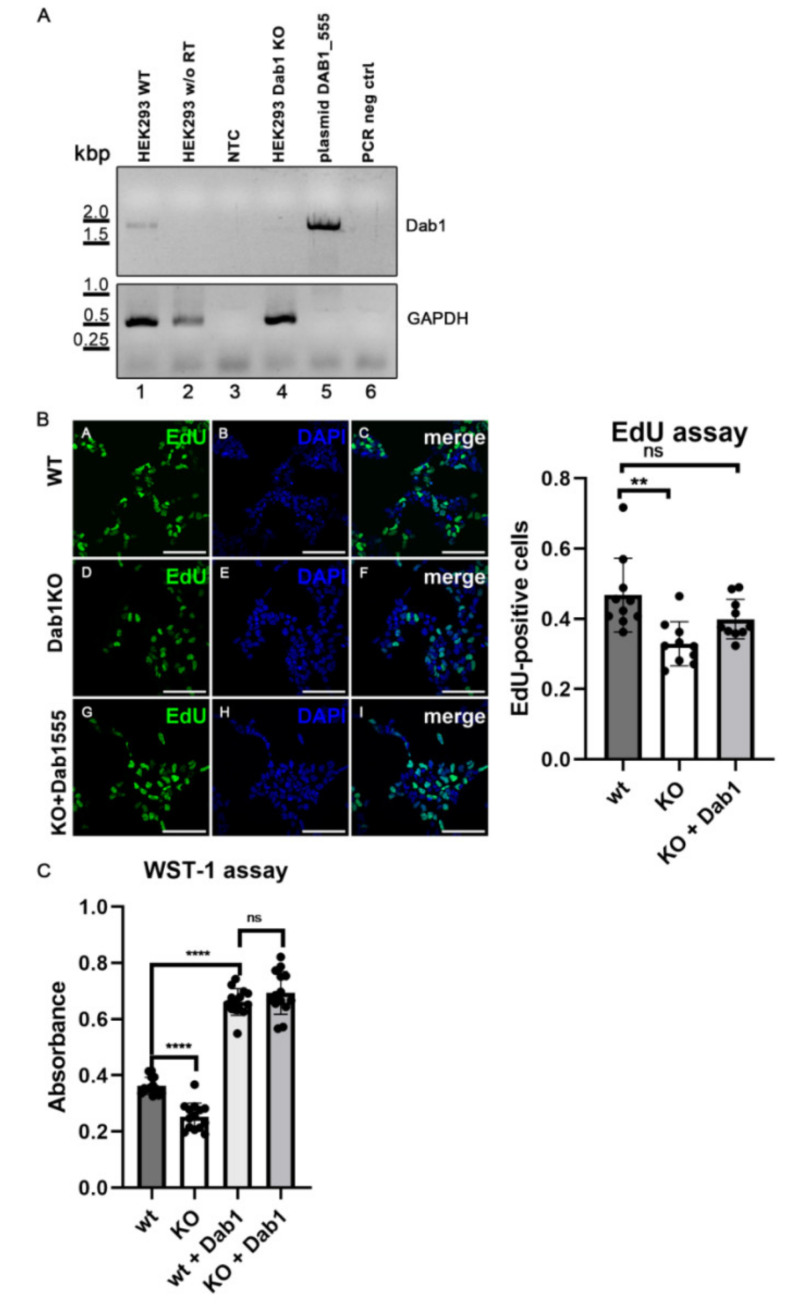
Cell proliferation rate is decreased in Dab1 k.o. cells. (**A**) Validation of HEK293 Dab1 k.o. cells. RNA from HEK293 wt and HEK293 Dab1 k.o. cells was extracted and reverse transcribed using the RevertAid H Minus First Strand cDNA Synthesis Kit (Thermo Scientific) using oligo (dT)_18_ primers. Subsequently, PCR using primers specific for complete cDNA sequence of human Dab1 was performed and the presence of the band with the expected size of about 1700 bp corresponding to human Dab1 (Lane 1) was analyzed by agarose gel electrophoresis. As positive control a plasmid carrying the Dab1 sequence was used (Lane 5). Negative controls used in this study: HEK293 w/o RT, no reverse transcriptase (Lane 2); NTC, no RNA template (Lane 3); PCR neg, no cDNA template (Lane 6). (**B**) HEK293 wt, Dab1 k.o. cells, and Dab1 k.o. cells expressing Dab1_555 were treated with EdU (10 µM) and incubated for 2 h at 37 °C. Cells were fixed and EdU incorporated DNA was stained via Click-iT assay with AlexaFluor488 and subsequently with DAPI. Pictures of 10 different fields of view were taken and analyzed by Image J. Scale bar represents 100 µm. EdU and DAPI stained nuclei were counted and set into relation. Data were analyzed by one-way ANOVA multiple comparison test. (**C**) Proliferation/survival WST-1 assay performed in HEK293 wt and Dab1 k.o. cells transfected with pClneo or Dab1_555. WST-1 (10 nM) was added to each well and absorbance (480/30 nm) was measured after 2 h. The data were derived from two independent experiments, which were performed with 7 replicates. Data were analyzed by one-way ANOVA multiple comparison test, ** *p* ≤ 0.01, **** *p* ≤ 0.0001, ns, not significant, dots, number of experiments.

**Figure 7 ijms-22-01745-f007:**
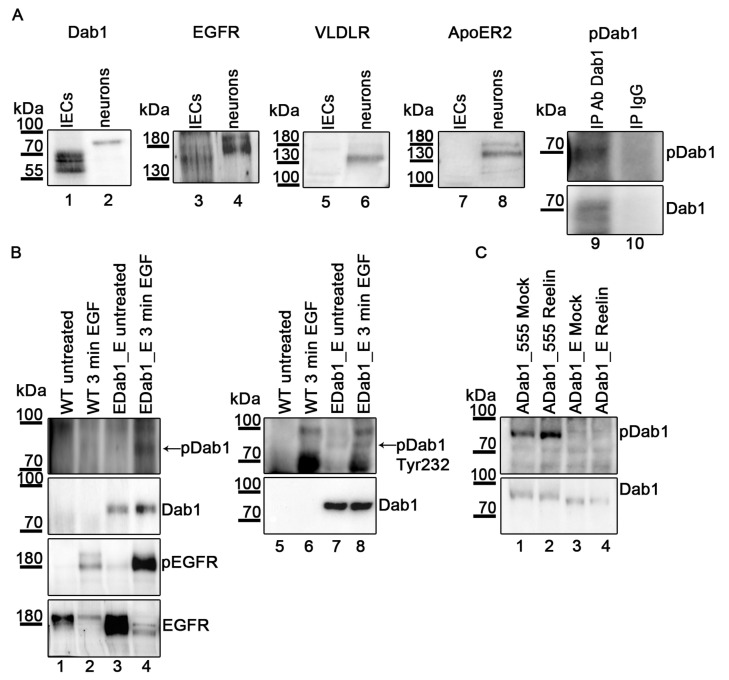
EGF induces phosphorylation of the Dab1 Early isoform. (**A**) Embryonic primary neurons (DIV3) and intestinal epithelial cells (IECs) were lysed and the protein extracts were analyzed by western blotting for expression of Dab1 (Lanes 1 and 2), EGFR (Lanes 3 and 4), VLDLR (Lanes 5 and 6), and ApoER2 (Lanes 7 and 8). Phosphorylation level of Dab1 in IECs was analyzed by Dab1 immunoprecipitation and probing eluates samples with phosphotyrosine specific antibody (Lanes 9 and 10, Panel 1). Blot was stripped and re-probed with a Dab1 specific antibody (Lanes 9 and 10, Panel 2). As negative control unrelated mouse IgG was used. (**B**) Western blot analysis of HEK293 cells (Lanes 1 and 2) and HEK293 cells overexpressing EGFR and chDab1_E (Lanes 3 and 4). Cells were either left untreated (Lanes 1 and 3) or treated with human EGF (10 ng/mL) for 3 min (Lanes 2 and 4). Immunoprecipitation using a Dab1 specific antibody (Ab 54) was performed. Dab1 phosphorylation levels (Panel 1, Lanes 1–4) were detected using an antibody against phosphorylated tyrosine residues (Ab PY99). The blot was stripped and re-probed for Dab1 using Ab D4 (Panel 2). Extracts were blotted with an antibody specific for EGFR phosphorylation (Panel 3, Lanes 1–4) and total levels of EGFR (Panel 4, Lanes 1–4). Extracts were also blotted with an antibody specific for Dab1 phosphorylation (Tyr 232) (Panel 1, Lanes 5–8). The blot was stripped and then re-probed with a Dab1 specific antibody (Panel 2, Lanes 5–8). (**C**) Western blot analysis of HEK293 cells expressing ApoER2 and Dab1_555 (Lanes 1 and 2) or chDab1_E (Lanes 3 and 4). Cells were either treated with Reelin (Lanes 2 and 4) or Mock conditioned medium (Lanes 1 and 3) for 20 min. Immunoprecipitation using a Dab1 specific antibody was performed (Ab 54). Dab1 phosphorylation levels (Panel 1) were detected using an antibody against phosphorylated tyrosine residues (Ab PY99). The blot was stripped and re-probed for Dab1 using Ab D4 (panel 2).

**Table 1 ijms-22-01745-t001:** The following antibodies were used in this study at the indicated dilutions.

Epitope	Catalog Number	Company/Reference	Dilution
pEGFR	sc-12351	Santa Cruz Biotechnology (Dallas, TX, USA)	WB 1:1000
EGFR	sc-373746	Santa Cruz Biotechnology (Dallas, TX, USA)	WB 1:1000, IP
ApoER2	Ab186	Produced in the lab	IP
ApoER2	Ab 20	Produced in the lab	WB 1:5000
VLDLR	AF2258	R&D Systems (Minneapolis, MN, USA)	WB 1:1000
p-Tyr (PY99)	sc-7020	Santa Cruz Biotechnology (Dallas, TX, USA)	WB 1:500
Phospho-Dab1 (Tyr232) Antibody	#3325	Cell Signaling Technology (Danvers, MA, USA)	WB 1:500
D4 Dab1		Kind gift from André Goffinet (UCLouvain)	WB 1:1000 NIH3T3 WB 1:8000 HEK293
GAPDH	G8795	Sigma-Aldrich (St. Louis, MO, USA)	WB 1:10,000
NeuN	Clone A60, MAB377	Merck Millipore (Burlington, MA, USA)	WB 1:1000
Normal rabbit IgG	12-370	Merck Millipore (Burlington, MA, USA)	IP
Normal mouse IgG	sc-2025	Santa Cruz Biotechnology (Dallas, TX, USA)	IP

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
