# Peer review of "Disabled 1 Is Part of a Signaling Pathway Activated by Epidermal Growth Factor Receptor"

_ijms, 2021, doi:10.3390/ijms22041745_

Round 1

Reviewer 1 Report

Dlugosz and colleagues describe a novel Reelin-independent signaling pathway involving the adapter protein Dab1, which is activated by the epidermal growth factor receptor. Their findings extend our knowledge on non-canonical Dab1 signaling and have potential important implications for EGFR-mediated cellular signaling, including oncogenic signaling.

Several control experiments / improvements are recommended:

In Figure 1C (coIP of Dab1 by EGFR antibody from embryonic brain lysate) the EGFR pulldown control band can barely be seen (panel 2), and the Dab1 blot (panel 1) is cut too close above the Dab1 band. The authors should provide, as in B for 293 cells, the control experiment with Apoer2 coimmunoprecipitating Dab1 from brain lysate. In D (panel M) the coimmunofluorescence of mCherry with DAb1-mGFP should be shown.

In Fig. 2E, the authors should demonstrate the effect of Reelin (RCM vs. MCM) on pEGFR in embryonic neurons. In addition the number of experiments performed should be specified and the standard error of –fold induction of pDab1 be indicated.

Since prolonged exposure of neurons to Reelin induces Dab1 degradation, it would be important to know if (or if not) this is also the case for EGFR-mediated Dab1 signaling in 293 or NIH3T3 cells. Did the authors perform this experiment ?

As Src kinases can phosphorylate Dab1, it would be important to measure Src kinase family activity and their potential inhibition by the EGFR inhibitors, which are used at concentrations of 1 μM, in Fig. 3. Ideally, the authors might use a complementary approach to block EGFR activation, either by using an inhibitory antibody, or by transfection of a catalytically inactive EGFR mutant. This would greatly strengthen their conclusion that EGF/EGFR directly phosphoryates Dab1.

In Figure 5C, the blots in panels 1 and 2 are cut too close above the Dab1 bands.

In Figure 7A, immunoprecipitation of Dab1 followed by incubation with the anti-pY blot would strengthen their conclusion that Dab1 is phosphorylated in intestinal epithelial cells. In 7B, please explain the additional bands below 70 and at around 100 kDa in lanes 2 and 4. Again, a Dab1 IP followed by anti-pY-WB would be preferable. In Fig. 7B and C, also show the EGFR phosphorylation.

Additional remarks:

The authors should cite their own work by Strasser et al. 2004 when discussing the role of receptor clustering by Reelin on Dab1 activation (pages 1 and 11).

In the Methods section, the preparation of embryonic neurons is described in detail, while the preparation of intestinal epithelial cells is not described at all. Please provide a brief description.

Author Response

Reviewer 1

Dlugosz and colleagues describe a novel Reelin-independent signaling pathway involving the adapter protein Dab1, which is activated by the epidermal growth factor receptor. Their findings extend our knowledge on non-canonical Dab1 signaling and have potential important implications for EGFR-mediated cellular signaling, including oncogenic signaling.

Several control experiments / improvements are recommended:

In Figure 1C (coIP of Dab1 by EGFR antibody from embryonic brain lysate) the EGFR pulldown control band can barely be seen (panel 2), and the Dab1 blot (panel 1) is cut too close above the Dab1 band. The authors should provide, as in B for 293 cells, the control experiment with Apoer2 coimmunoprecipitating Dab1 from brain lysate. In D (panel M) the coimmunofluorescence of mCherry with DAb1-mGFP should be shown.

Co-IP of Dab1 by ApoER2 was performed as suggested and the results presented in Fig. 1C (lanes 1 – 3).

Co-IP of Dab1 by EGFR from embryonic neurons was re-done (Fig. 1C lane 5) and the whole blot was cut further away from the Dab1 band.

Co-immunofluorescence of mCherry with Dab1-mGFP is now shown in Fig. 1D (panels M – O)

In Fig. 2E, the authors should demonstrate the effect of Reelin (RCM vs. MCM) on pEGFR in embryonic neurons. In addition the number of experiments performed should be specified and the standard error of –fold induction of pDab1 be indicated.

The effect of Reelin on EGFR phosphorylation is now demonstrated (expansion of Fig. 2E) and the significance of the effect on Dab1 phosphorylation was evaluated and shown as additional panel in Fig. 2E.

Since prolonged exposure of neurons to Reelin induces Dab1 degradation, it would be important to know if (or if not) this is also the case for EGFR-mediated Dab1 signaling in 293 or NIH3T3 cells. Did the authors perform this experiment?

We performed this experiment and 1h after the addition of EGF there was no decrease in Dab1, probably because of Dab1 overexpression in these cells lines. That’s why we did not add it to the manuscript.

As Src kinases can phosphorylate Dab1, it would be important to measure Src kinase family activity and their potential inhibition by the EGFR inhibitors, which are used at concentrations of 1 μM, in Fig. 3. Ideally, the authors might use a complementary approach to block EGFR activation, either by using an inhibitory antibody, or by transfection of a catalytically inactive EGFR mutant. This would greatly strengthen their conclusion that EGF/EGFR directly phosphoryates Dab1.

We transfected HEK293T cells with plasmids expressing Dab1 and either EGFR WT or EGFR KD (Kinase dead, K721A). Cells were treated with EGF or left untreated. Results from this experiment are now present in Fig.3C.

In Figure 5C, the blots in panels 1 and 2 are cut too close above the Dab1 bands.

We redid the whole experiment with primary neurons from ApoER2-/-/VLDLR-/- mice and unfortunately the results were not conclusive. Thus we decided to remove panel C in Fig. 5. For the time being we have to leave it open whether the EGF/EFGR induced Dab1 phosphorylation increases in neurons when both receptors are absent.

In Figure 7A, immunoprecipitation of Dab1 followed by incubation with the anti-pY blot would strengthen their conclusion that Dab1 is phosphorylated in intestinal epithelial cells. In 7B, please explain the additional bands below 70 and at around 100 kDa in lanes 2 and 4. Again, a Dab1 IP followed by anti-pY-WB would be preferable. In Fig. 7B and C, also show the EGFR phosphorylation.

To test for the presence of phosphorylated Dab1 in intestinal cells we performed the experiment suggested by the reviewer. Dab1 was precipitated and the precipitate was probed with an anti-pY antibody and re-probed with D4 for Dab1. The results of this experiment are shown now as Fig. 7A lanes 9 and 10 which replace former lane 9.

Evaluation of Dab1E phosphorylation by EGF in HEK293 cells was done by Dab1 IP followed by anti-pY-WB as suggested.

In the new Fig. 7B phosphorylation of EGFR is also shown.

Additional remarks:

The authors should cite their own work by Strasser et al. 2004 when discussing the role of receptor clustering by Reelin on Dab1 activation (pages 1 and 11).

Our paper (Strasser et al., 2004) is now being cited in the corresponding sections of the Introduction and the Discussion

In the Methods section, the preparation of embryonic neurons is described in detail, while the preparation of intestinal epithelial cells is not described at all. Please provide a brief description.

A detailed description of the preparation of intestinal epithelial cells was added to the Methods section

Reviewer 2 Report

The manuscript by Dlugosz et al presents a novel demonstration that the Dab1 adaptor protein, which is typically associated with Reelin-Dab1 signaling also acts downstream of the EGF receptor in some circumstances. They demonstrated binding of Dab1 with the EGF receptor and phosphorylation downstream of EGF addition. Importantly they show that coexpression of the Reelin receptor ApoER2 competes with EGFR and prevents phosphorylation of Dab1 in response to EGF when co-expressed with EGFR. This provides a understanding of a hierarchy of pathways where the receptors are co-expressed.

While this is a high quality study, I see areas that can be improved that could improve the impact of the manuscript.

In the introduction the connection between the Reelin signaling pathway and AKT is documented but the arguably more important downstream signaling to C3G/Rap1/N-cadherin was ignored. The consensus of high affinity Dab1 PTB domain binding extends outside of the NPxY motif to upstream to a larger domain Y/FxNPxY. This was not mentioned and it may explain the ability of ApoER2 to compete with EGFR for binding. Several transmembrane receptors contain the NPxY motif including several ApoE and growth factor receptors. It would seem that the NPxY motif on its own is not sufficient for binding and connecting Dab1 to a signaling pathway. This could be a discussion point.

In the result section (line 86) it is not clear why exclusion from the nucleus of tagged proteins is discussed as proof that the tags don't contribute to the finding. Exclusion from the nucleus of the fusion protein could be a consequence of its larger size.

Later (l162) it is stated that "we have to assume that the mechanism by which Dab1 is phosphorylated by EGFR is different than that mediated by ApoER2..." ApoER2 leads to increased Dab1 phosphorylation through a partially understood mechanism that involves the activation of Src. The EGFR also activates Src. It was not tested in this study whether Src inhibitors are able to inhibit the phosphorylation of Dab1 in response to EGF stimulation. The authors are careful around this point always referring to Dab1 phosphorylation as a response to EGR stimulation instead of indicating that Dab1 is phosphorylated by EGFR.

Curiously the authors used serum instead of EGF to stimulate cell proliferation of wild-type and dab1 KO HEK293 cells (Fig. 6). This weakens their ability to put Dab1 on the EGF pathway in the regulation of cell proliferation since serum contains several mitogens, but clearly connects Dab1 to proliferation.  Cell proliferation was not examined in the intestinal epithelial cells, although it was shown that EGF is capable of inducing Dab1 phosphorylation. The abstract appears to be overstated on these points making the claim (l15) "This pathway is involved in cell proliferation and/or cell survival and might contribute to ..." This should be either fixed experimentally or written in a manner that does not overstate the finding.

Author Response

Reviewer 2

The manuscript by Dlugosz et al presents a novel demonstration that the Dab1 adaptor protein, which is typically associated with Reelin-Dab1 signaling also acts downstream of the EGF receptor in some circumstances. They demonstrated binding of Dab1 with the EGF receptor and phosphorylation downstream of EGF addition. Importantly they show that coexpression of the Reelin receptor ApoER2 competes with EGFR and prevents phosphorylation of Dab1 in response to EGF when co-expressed with EGFR. This provides a understanding of a hierarchy of pathways where the receptors are co-expressed.

While this is a high quality study, I see areas that can be improved that could improve the impact of the manuscript.

In the introduction the connection between the Reelin signaling pathway and AKT is documented but the arguably more important downstream signaling to C3G/Rap1/N-cadherin was ignored. The consensus of high affinity Dab1 PTB domain binding extends outside of the NPxY motif to upstream to a larger domain Y/FxNPxY. This was not mentioned and it may explain the ability of ApoER2 to compete with EGFR for binding. Several transmembrane receptors contain the NPxY motif including several ApoE and growth factor receptors. It would seem that the NPxY motif on its own is not sufficient for binding and connecting Dab1 to a signaling pathway. This could be a discussion point.

The C3G/Rap1/N-cadherin aspect of the Reelin pathway was added to the Introduction

The remark that the binding site for Dab1 extends outside of the NPxY motif in the receptor was highly appreciated. We have added this aspect to the discussion and used this argument to better explain our findings.

In the result section (line 86) it is not clear why exclusion from the nucleus of tagged proteins is discussed as proof that the tags don't contribute to the finding. Exclusion from the nucleus of the fusion protein could be a consequence of its larger size.

We have remodeled (additional controls) Fig. 1D according to the suggestion of reviewer 1 and have deleted our thoughts about exclusion of the tagged proteins from the nucleus.

Later (l162) it is stated that "we have to assume that the mechanism by which Dab1 is phosphorylated by EGFR is different than that mediated by ApoER2..." ApoER2 leads to increased Dab1 phosphorylation through a partially understood mechanism that involves the activation of Src. The EGFR also activates Src. It was not tested in this study whether Src inhibitors are able to inhibit the phosphorylation of Dab1 in response to EGF stimulation. The authors are careful around this point always referring to Dab1 phosphorylation as a response to EGR stimulation instead of indicating that Dab1 is phosphorylated by EGFR.

We tested the Src inhibitor PP2 in HEK293 cells expressing ApoER2/EGFR, and Dab1 and treated the cells either with Reelin (positive control) or with EGF. Unfortunately addition of PP2 also decreased phosphorylation of EGFR demonstrating that it is not as specific as originally thought, see also: https://www.ncbi.nlm.nih.gov/pmc/articles/PMC3423592/

To further prove that Dab1 phosphorylation depends on the kinase avtivity of EGFR we have expressed a kinase dead version of EGFR. This variant of the receptor was not able to phosphorylate Dab1 upon EGF treatment. The results of this additional experiment are presented as Fig. 3C in the revised manuscript.

Curiously the authors used serum instead of EGF to stimulate cell proliferation of wild-type and dab1 KO HEK293 cells (Fig. 6). This weakens their ability to put Dab1 on the EGF pathway in the regulation of cell proliferation since serum contains several mitogens, but clearly connects Dab1 to proliferation.  Cell proliferation was not examined in the intestinal epithelial cells, although it was shown that EGF is capable of inducing Dab1 phosphorylation. The abstract appears to be overstated on these points making the claim (l15) "This pathway is involved in cell proliferation and/or cell survival and might contribute to ..." This should be either fixed experimentally or written in a manner that does not overstate the finding.

Due to severe time restrains (7 days) we were not able to fix this experimentally and thus, we deleted this this statement in the abstract as suggested by the reviewer.

Round 2

Reviewer 1 Report

The authors have addressed both reviewers' comments and suggestions in an adequate and timely manner. It is a pity that the results from Apoer2/Vldlr dKO neurons were inconclusive, but given the short period of time available for revision it is adequate to remove this part from the revised manuscript.